# Leucine and Arginine Availability Modulate Mouse Embryonic Stem Cell Proliferation and Metabolism

**DOI:** 10.3390/ijms232214286

**Published:** 2022-11-18

**Authors:** Bibiana Correia, Maria Inês Sousa, Ana Filipa Branco, Ana Sofia Rodrigues, João Ramalho-Santos

**Affiliations:** 1Department of Life Sciences, University of Coimbra, Calçada Martim de Freitas, 3000-456 Coimbra, Portugal; 2CNC—Center for Neuroscience and Cell Biology, CIBB, University of Coimbra, Azinhaga de Santa Comba, Polo 3, 3000-354 Coimbra, Portugal

**Keywords:** Leucine, Arginine, mESC

## Abstract

Amino acids are crucial nutrients involved in several cellular and physiological processes, including fertilization and early embryo development. In particular, Leucine and Arginine have been shown to stimulate implantation, as lack of both in a blastocyst culture system is able to induce a dormant state in embryos. The aim of this work was to evaluate the effects of Leucine and Arginine withdrawal on pluripotent mouse embryonic stem cell status, notably, their growth, self-renewal, as well as glycolytic and oxidative metabolism. Our results show that the absence of both Leucine and Arginine does not affect mouse embryonic stem cell pluripotency, while reducing cell proliferation through cell-cycle arrest. Importantly, these effects are not related to Leukemia Inhibitory Factor (LIF) and are reversible when both amino acids are reconstituted in the culture media. Moreover, a lack of these amino acids is related to a reduction in glycolytic and oxidative metabolism and decreased protein translation in mouse embryonic stem cells (mESCs), while maintaining their pluripotent status.

## 1. Introduction

Nutrients have been previously shown to be important instruments for initial embryonic development, and a lack of specific components in specific phases of the process severely jeopardizes correct development, including developmental interruption [1]. It is clear that the physical and chemical support provided by maternal tissues has a relevant role for initial embryogenesis and implantation, and it may even affect fetal development and postnatal life [2,3,4]. In fact, the embryo’s journey from the oviduct to the uterus involves many changes to surrounding environmental conditions. In this regard, amino acid availability, as well as oxygen tensions, are very important in influencing embryo metabolism and redox state, promoting survival and morphological changes, adaptation to stress, regulating epigenetic and transcriptional events, signaling and biosynthetic pathways, and thus modulating development potential [2,3,4,5,6].

Amino acids have different cellular roles, ranging from stress adaptation, intracellular osmolarity maintenance, protein synthesis, modulation of signaling, and especially maintaining the cytosolic NAD+ pool, which in turn is essential for cellular metabolic flow [5,7,8]. Decreased amino acid availability, especially branched chain amino acids such as Leucine (Leu), occurs in the surrounding embryo environment as a consequence of low maternal protein consumption [3]. Furthermore, Leucine (Leu) and Arginine (Arg) withdrawal from media has been shown to negatively regulate trophectoderm cell outgrowth in the conceptuses of different species [9,10,11,12,13,14,15,16], as well as cell migration and proliferation [9,12,16,17]. Consequently, blastocysts cultured under these conditions remained in a reversible developmental quiescent state [9,12,16].

However, previous work has focused on trophectoderm cells, not on the effects that Leu and Arg absence might have on the inner cell mass (ICM) and therefore on ESCs, which are the cells destined to form embryonic tissues. Therefore, the main aim of this study was to explore whether Leu and Arg withdrawal from pluripotent mouse embryonic stem cell (mESCs) culture medium has an effect on the growth and self-renewal of these cells and how these amino acids interfere with cellular processes on a metabolic level. In this work, we report the results of Leu and Arg withdrawal on the properties of mouse embryonic stem cells.

## 2. Results

### 2.1. Leucine and Arginine Withdrawal Regulates mESC Self-Renewal and Viability

Leu and Arg withdrawal from embryo culture has previously been shown to affect embryo outgrowth, and consequently its development at the blastocyst stage [9,10,11,12,13,14,15,16,17]. However, trophectoderm was the focus of these studies, which never addressed the effects of Leu and Arg absence on embryonic pluripotent stem cells. In order to address this issue, we cultured mESCs in customized 2i media formulated without both Leu and Arginie (xAA condition). Interestingly, we observed that mESC colonies could be maintained in culture without both amino acids for at least 20 days, despite a progressive reduction in size and colony number (Figure 1a). Additionally, we assessed the reversibility of this effect, and for this purpose, after 19 days in culture, we replaced the culture medium with regular 2i media and observed that even after 19 days of Leu and Arg starvation, cells restarted growing, increasing colony size. In fact, after 5 days, we detected expanded colonies, which, after passage, continued to display the normal morphology of 2i mESCs in culture (Figure 1a). To understand whether these cells retained pluripotency, we performed immunostaining for NANOG, showing that cells were positive for this pluripotency marker (Figure 1b). We then sought to further explore the growth behavior of these cells. We found that cell proliferation under the xAA condition was clearly slower when compared to a regular culture in 2i/LIF. Cells were to be maintained in culture and counted for at least 23 days, when we decided to end the experiment (Figure 1c). It is noteworthy that 23 days is longer than the time of a murine pregnancy. Taking everything into account, these results led us to perform a growth curve experiment until the 48 h time point. As expected, cells cultured in 2i/LIF displayed exponential growth (Figure 1d). In contrast, proliferation of cells in the xAA condition was clearly slower, and within 24 h, the cell number was significantly decreased compared to the other 2i/LIF, which was also observed at 48 h (Figure 1d). To rule out any potential cytotoxic effects, viability assays were performed, since effects observed in proliferation could also be explained by an increase in cell death. Therefore, after 48 h under experimental conditions, we performed an Annexin V/PI apoptosis assessment assay and observed that there were no differences between the percentage of viable cells in conditions, either for early and late apoptosis or necrosis (Figure 1e). Accordingly, we also evaluated whether xAA cultured cells were able to recover cell numbers after the 2i/LIF medium without Leu and Arg was supplemented with both Leu and Arg, matching the final concentration found in DMEM-F12/Neurobasal media (82 mg/L and 115.75 mg/L, respectively). Although cells previously cultured in xAA started to increase in number, as a result of the discrepancy in the number of cells between conditions when the recovery period started, this is not perceived when observing the growth curve (Figure 1f). For this reason, we calculated the growth rates for each condition. With a decreased growth rate, the cells become less proliferative; in fact, it is evident that at 48 h, xAA cells displayed significantly lower growth rates compared to those from 2i/LIF (Figure 1g). Furthermore, after adding Leu and Arg to mESCs cultured for 48 h in their absence, at both time points of the recovery period, there was an increase in cell numbers in the xAA condition (Figure 1f), which is more notorious considering that the increase in growth rate (Figure 1g) is significantly different from 2i/LIF after 96 h. On the other hand, the decrease in growth rate in 2i/LIF reflected the overpopulated culture (Figure 1f,g), with limited space available in the culture dishes to allow growth. Given these effects on proliferation, we then sought to understand whether amino acid absence may be affecting the cell cycle. In fact, after 24 h, the percentage of xAA cultured cells in G1 phase was significantly higher when compared to cells in 2i/LIF (Figure 1h), and this difference increased after 48 h (Figure 1i). Concomitantly, this alteration is accompanied by a significant decrease in the percentage of cells in the S and G2/M phases of the cell cycle (Figure 1g,h). When assessing the cell-cycle distribution of mESCs during the recovery period, we observed an increase in cells in the S and G2/M phases at both 72 h and 96 h (Figure 1i,j), indicating progression through the cell cycle following Leu and Arg reconstitution back into the culture medium. Thus, a lack of Leu and Arg seems to affect mESC proliferation via the cell cycle, interfering with progression from G1 to the S phase and G2/mitotic phases.

### 2.2. Effects of LIF Depletion along with Leucine and Arginine on mESC Self-Renewal, Cell Cycle, and Viability

LIF is a soluble cytokine, important in naïve embryonic stem-cell culture [18,19]. Additionally, LIF is synthesized by inner cell-mass cells (embryonic stem cells within the blastocyst) and uterine tissues, promoting uterine receptivity and embryo implantation [20,21]. Moreover, the absence of LIF affects embryo outgrowth and further development, despite not being required for embryo viability [20,21,22]. Taking into consideration these roles of LIF, we also aimed to evaluate whether the absence of LIF, in addition to Leu and Arg deprivation, would have the same impact on mESC proliferation as the absence of only Leu and Arg. In order to evaluate the impact of Leu and Arg and LIF deprivation on culture proliferation, we cultured cells in regular 2i medium without LIF supplementation (xLIF condition) and in xAA medium without LIF supplementation (xAAxLIF condition). Similar to the effects observed in the xAA growth curve, cells in xAAxLIF proliferated significantly less than cells in 2i/LIF or xLIF (Figure 2a), with irrelevant cell death (Figure 2b). Moreover, these cells can also be cultured for at least 23 days (Figure 2c), much like cells cultured in xAA medium. On the contrary, cells cultured without LIF presented an exponential growth similar to those cultured in 2i/LIF (Figure 2a). In addition, during recovery, the number of cells in the xAAxLIF condition increases (Figure 2a), which is more evident when looking at the growth rates. Indeed, the significant differences in the T = 72 h–T = 96 h interval, between the increased growth rate in the xAAxLIF condition compared to 2i/LIF and xLIF, show that cells recover exponential growth (Figure 2a,d). Furthermore, the decreased growth rate of xLIF cells showed the same trend as 2i/LIF (Figure 2d), and, after 96 h, the culture was also overpopulated. In agreement with the results for proliferation, after 24 h (Figure 2e) and 48 h (Figure 2f), the percentage of cells in the G1 phase was significantly higher in xAAxLIF cultured cells when compared to those in 2i/LIF and xLIF. Simultaneously, this alteration is accompanied by a significant decrease in the percentage of cells in the S and G2/M phases (Figure 2e,f). Additionally, xLIF cells, in accordance with the proliferation results, present an equivalent outcome at both time points as the control 2i/LIF condition (Figure 2e,f). Likewise, a lack of LIF, Leu, and Arg already affects progression from the G1 phase to the S and G2/mitotic phases after a 24 h culture period (Figure 2e), and the percentage of cells in the G1 phase is higher at 48 h (Figure 2f). However, analyzing the cell-cycle distribution of xLIF cells, the similarity to the 2i/LIF culture is noticeable (Figure 2e,f), even during recovery, when LIF is supplemented into the medium. Indeed, the percentage of cells in each phase of the cell cycle, at 72 h (Figure 2g) and also at 96 h (Figure 2g), remains identical to those of 2i/LIF. For this reason, we decided to end the experiments with the xLIF condition. When assessing the cell-cycle distribution of naïve mESCs of the xAAxLIF condition during the recovery period, we observed a decline in cells in G1, at the same time that the percentage of cells in the S and G2/M phases at both 72 h (Figure 1g) and 96 h (Figure 1h) increased, indicating progression through the cell cycle following LIF, Leu, and Arg reconstitution back into the culture medium. It is noteworthy that these results, in accordance with the growth curve and growth rate data, stress the notion that while Leu and Arg withdrawal (with LIF removal) induces proliferation arrest, caused by an impairment of progression from the G1 phase, LIF removal alone seemingly has no effect. This may occur because cells achieve a proliferative plateau, consistent with proliferation data, due to overpopulation.

### 2.3. Leucine and Arginine Withdrawal Does Not Affect Pluripotency

The effects of Leu and Arg withdrawal on mESC proliferation and the cell cycle are remarkable and in agreement with that which was observed for the trophectoderm cells and blastocysts when these amino acids are depleted from culture media [10,15,17]. Nevertheless, it is important to note that this does not negatively impact the cellular identity of naïve mESCs, nor their developmental potential. Regarding OCT4 (octamer-binding transcription factor 4 or POU domain, class 5, transcription factor 1) localization, this protein is detected at the nucleus (Figure 3a) in all conditions, as expected. Subsequently, we assessed the protein and mRNA levels of core pluripotency markers as well as markers of the pluripotency circuitry. Results demonstrate that protein expression of the core pluripotency genes OCT4, NANOG, and SOX2 (SRY-box transcription factor 2) is identical between conditions (Figure 3b). However, the levels of c-MYC (cellular myelocytomatosis oncogene) expression are lower when Leu, Arg, and LIF are absent in cell culture (Figure 3b). In terms of mRNA levels, for *Oct4*, *Nanog*, or the pluripotency circuitry genes *Esrrb* (estrogen related receptor beta) and *Rexo1* (RNA exonuclease 1), no significant effects on the expression of these genes were observed in any of the conditions (Figure 3d). We also evaluated the differentiation potential of cells in each condition to confirm that these cells can contribute to all cell lineages, without any bias towards any specific embryonic leaflet. For this purpose, an embryoid body (EB) assay was carried out, and markers for each cell lineage were analyzed by RT-PCR and immunostaining. Despite the decrease in c-MYC protein levels for xAA and xAAxLIF cells, the potential of these cells to differentiate was not affected, since during the 15 days of the assay, cells were able to produce embryoid bodies that not only shared morphological similarities (Figure 3e) but also adhered, expanded, and differentiated similarly. This is shown by the uniformity of mRNA expression levels in each condition of *aFp* (alpha fetoprotein- endoderm layer marker) and *b3Tub* (beta-3-tubulin- ectoderm layer marker). Despite the trend to an increased *Sma* (smooth muscle actin- mesoderm layer marker) expression in xAA, there was no statistical significance between the expression levels of each condition (Figure 3f). Additionally, we were also able to detect positive staining in cells of each condition for the protein presence of SMA by immunocytochemistry (Figure 3g), b3TUB (Figure 3h), and AFP (Figure 3i). Overall, our results suggest that pluripotency is not affected by withdrawing Leu and Arg from the culture medium or by LIF withdrawal in addition to Leu and Arg removal.

### 2.4. Lack of Leucine and Arginine Downregulates Glycolytic and Oxidative Function in mESCs as well as Overall Translation Initiation

Amino acid availability plays different roles in stress adaptation, signaling and gene expression, protein synthesis, and cellular metabolic flow [5,7,8]. Given the ambivalent metabolic nature of naïve ESCs towards both glycolysis and oxidative metabolism—meaning they are able to adapt and use different available substrates [23,24,25]—we further analyzed the effects of amino acid absence on these metabolic processes. Thus, we evaluated whether a lack of Leu and Arg influenced the consumption and production of key metabolites present in the extracellular media, such as glucose, pyruvate, and lactate. Although glucose is similarly internalized by cells in all conditions (Figure 4a), there is a trend for a slightly lower uptake of glucose in an amino acid absence. On the other hand, pyruvate uptake is significantly affected when both amino acids are absent (Figure 4a). The same is true for lactate production, which was higher in 2i/LIF cultured cells than in cells grown with no Leu and Arg (Figure 4a), indicating that glycolytic function is diminished. To further understand how these differences translate to glycolytic function, the extracellular acidification rates (ECAR) were measured after 48 h in the different treatment conditions using the Seahorse XF Analyzer. Under basal conditions, the ECAR of xAA and xAAxLIF cells was remarkably lower than thot observed for cells in 2i/LIF (Figure 4b). Consistent with the extracellular metabolite quantification results, cells in each condition reacted to glucose injection by increasing ECAR levels (Figure 4b); however, xAA and xAAxLIF cells are less responsive to glucose than cells in 2i/LIF (Figure 4b) given the significantly decreased glucose consumption, and these differences were more pronounced in the xAAxLIF condition. The oligomycin response in xAA and xAAxLIF cells is also significantly diminished compared to the control. This difference is also exacerbated in xAAxLIF (Figure 4b), indicating that the energy demand for ATP is lower. To determine whether the decrease in glycolytic function could be explained by changes in the expression of some key metabolic enzymes, we evaluated the mRNA levels of *Hxk2* (hexokinase 2) and *Pkm2* (pyruvate kinase 2), which catalyze irreversible reactions in glycolysis, as well as *Ldha* (Lactate dehydrogenase A). Despite the lack of statistically significant differences in *Hxk2* and *PKm2* expression levels (Figure 4c), the expression of the *Ldha* gene was downregulated when both amino acids were absent (Figure 4c). Concomitantly, the protein levels of PKm2 and HXK2 were not significantly affected (Figure 4d,e), but a decrease in LDHA protein expression was noticeable in the xAA and xAAxLIF conditions LIF (Figure 4d,e). Additionally, these differences were more evident in the case of the LDHA phosphorylated form and the ratio between the phosphorylated and total form of this protein. Given that pyruvate consumption was affected upon Leu and Arg withdrawal (Figure 4a) and that oligomycin disruption of the ATP production in the mitochondria in the glycolytic assay only mildly accelerated the glycolysis rate, we also performed a cell respiration oxidative assay using the Seahorse XF Analyzer. This assay monitors oxygen consumption rates (OCR) in order to infer putative effects on mitochondrial function. The global oxidative function of cells cultured without Leu and Arg was decreased in contrast to the global oxidative pattern of 2i/LIF cultured cells (Figure 4f). This is seen under basal conditions, with significantly lower basal respiration rates (Figure 4g), and reduced the levels of respiration associated with ATP production (Figure 4g). Although no differences were found in maximal oxygen consumption (Figure 4g), we observed a tendency towards a decrease when both amino acids were absent, which is in agreement with the difference in OCR when FCCP (Carbonyl cyanide 4-trifluoromethoxy phenylhydrazone) is injected, mimicking an energetic crisis (Figure 4f). This implies that xAA cells have lower metabolic demands to rapidly oxidize substrates; however, these are still functional when required. Although there were no differences in the expression of crucial proteins for oxidative phosphorylation (OXPHOS) as SDHA (succinate dehydrogenase a) or PDHK (pyruvate dehydrogenase kinase) (Figure 4h,i), we observed that xAA and xAAxLIF cells have significantly lower protein expression of COX4 (cytochrome c oxidase subunit 4), which is part of Complex IV of the mitochondrial respiratory chain (Figure 4h,i), essential for proton gradient maintenance and oxygen consumption by the mitochondrial respiratory chain. Regarding the data attained for energy metabolism and accounting for the fact that these pathways are also anaplerotic, these results led us to consider whether overall transcription and translation initiation are downregulated under Leu and Arg withdrawal. Surprisingly, there was a trend toward a decrease in transcriptional activity, but there were no significant differences when comparing all experimental conditions (Figure 4j). However, 2i/LIF cells display significant differences for transcription compared to a control using the inhibitor Actinomycin D (ACTD; Figure 4j), as expected. Furthermore, the initiation of the overall translation of xAA and xAAxLIF cells was as decreased as that of cells treated with cycloheximide (CHX), a specific translation inhibitor (Figure 4k), whereas the nascent protein synthesis level of 2i/LIF cultured cells is remarkably different from this positive control, as well as from xAA and xAAxLIF conditions, as expected. Taken together, our results support the notion that Leu and Arg absence reduces the intake of pyruvate and glucose consumption, promoting the downregulation of overall glycolytic and oxidative function and consequently decreasing the production of lactate, which is consistent with the less-active form of lactate dehydrogenase. Although no effects were seen on the overall nascent RNA synthesis level, the nascent protein synthesis is lower when both Leu and Arg are depleted from the mESC culture medium, indicating that this biosynthetic pathway may be downregulated.

## 3. Discussion

More than building blocks for the translation process, amino acids play important physiological roles indispensable for regulating cellular processes, from metabolic activity [5,6,26,27], to growth and proliferation [28,29,30], signaling pathways [5,31,32], gene expression [6,33], stress adaptation [5], and intracellular osmolarity maintenance [34]. As expected, amino acid balance is crucial for reproduction, from oocyte development to implantation and placentation as well as embryonic and fetal development [3,4]. Some studies focusing on amino acids such as threonine and glycine, among others [6], have shown, using mESCs as an in vitro model for ICM, that proliferation was slower [6]. On the other hand, Leu, an essential amino acid, and Arg, a semi-essential amino acid, have been shown to negatively regulate trophectoderm cell outgrowth in the conceptuses of different species [9,10,11,12,13,14,15,16]. Consequently, blastocysts remained in a reversible developmental quiescent state [9,12,16]. Leu and Arg depletion resulted in blastocyst developmental arrest, which suggests an embryo development arrest in vitro, resembling a diapause state. In parallel, Leu and Arg withdrawal was also explored in trophectoderm cells cultured in vitro, in which it promoted decreased nucleic acid synthesis [35,36] and other effects such as slow proliferation [10,37] and therefore ineffective invasion. However, these studies were mostly focused on trophectoderm cells in vivo and in vitro and trophoblast outgrowth [9,12,16,38], rather than ICM cells and embryonic stem cells, in which Leu and Arg effects are poorly studied.

In this work, we used naive mouse embryonic stem cells cultured in a 2i/LIF system as a starting point to predict how Leu and Arg absence would affect ICM cells. Our results demonstrate that Leu and Arg starvation was indeed able to reduce cell proliferation, affecting cell-cycle progression, causing an increase in cells in the G1 phase and a concomitant decrease in cells in the S or G2/M phases in a 24 h period. Furthermore, mESCs cultured in the absence of Leu and Arg are sustainable in culture for at least 23 days, without losing their recovery capacity and maintaining pluripotency markers. This indicates that mESCs cultured in the absence of these two amino acids remain quiescent, similar to that which was observed for trophectoderm cells. Moreover, annexin V/PI results show that amino acid withdrawal does not compromise cell viability. Interestingly, naive pluripotent stem cells are maintained in a 2i/LIF culture and LIF, a soluble cytokine fundamental for implantation and triggering embryo invasion [20,21,22]. Indeed, experimentally, it is possible to prevent embryo implantation in a LIF-dependent manner by ovariectomy of female mice, given that ovarian estrogen production regulates uterine receptivity, and in its absence, embryos arrest development (embryonic diapause) [20,21,22]. Therefore, female LIF-deficient mice generate viable embryos, but implantation only occurs when these embryos are transferred to a pseudo-pregnant recipient [20]. In addition, embryos lacking part of the LIF receptor heterodimer are unable to develop beyond the blastocyst stage [22], demonstrating the contribution of LIF signaling to this process. In light of such facts, we tried to understand whether the absence of LIF would be a limiting factor for the effects of Leu and Arg established in mESCs. In our culture system, the absence of LIF alone did not interfere in cell proliferation nor cell-cycle progression, in agreement with the previous literature [39,40,41]. Nonetheless, LIF, Leu, and Arg withdrawal were tested together to determine whether the cytokine was a limiting factor for either paused-pluripotency induction or its recovery. However, our data for cell proliferation, viability, and cell cycle demonstrate that this is not the case. The advantage of culturing mouse pluripotent stem cells with LIF comes from its role in promoting the expression of pluripotency circuitry genes, which are important for stabilizing the expression of Oct4, Nanog, and Sox2, the core pluripotency transcription factors [18,19,41,42,43]. Although the xAAxLIF condition lacked LIF in its final formulation, expression of these factors was not impaired in any of the experimental conditions. These results were expected, considering that the 2i medium is a robust system to maintain naive pluripotency in vitro by inhibiting both the MAPK/Erk pathway and GSK3β/Wnt42, not requiring the obligatory supplementation with LIF to sustain pluripotency. More importantly, the effects of both amino acid and LIF removal were reversible. It is also noteworthy that pluripotency was not affected either in Leu and Arg withdrawal or when LIF was not added to the culture. In fact, cells not only maintained the expression of pluripotency markers, but were also able to normally differentiate, expressing markers of the three embryonic leaflets.

Interestingly, decreased growth and proliferation and cell-cycle arrest, accompanied by slower rates of biosynthetic processes, such as those involved in DNA, RNA, and protein synthesis [35,37,44], are features of embryonic diapause. Diapause is a reproductive strategy that is geared towards delaying development if optimal environmental conditions are not met, thus preserving reproductive potential. Elective embryonic diapause (ED) is frequent in some mammals, such as mice [45,46]. Embryos initially develop from zygote to blastocyst, and at this stage, development is arrested, similarly to that which occurs when Leu and Arg are depleted from embryo culture conditions. These embryos are also characterized by a low metabolic profile [23,47,48] and a downregulation of metabolic-related genes [35,36]. Interestingly, a recent work reported that embryonic stem-cell amino acid starvation is able to induce a reversible paused-pluripotency state [26], with a parallel upregulation of glycolytic-related genes and glycolytic flux. This is reminiscent of the proliferation arrest that Leu and Arg depletion causes in cancer cells [49,50,51,52,53,54,55], as well as Arg removal in particular, which affects mitochondrial activity, by limiting OXPHOS through downregulation of OXPHOS-gene expression [53].

However, in this work, the absence of both amino acids promotes differential pyruvate uptake and glucose consumption, traduced by a lower glycolytic metabolism profile, confirmed by the extracellular acidification rates (ECAR) and additionally supported by decreased lactate secretion. This is not surprising, given the decreased *ldha* gene expression, as well as the concomitant protein levels, especially the notorious decrease in the phosphorylated (active) form. Interestingly, c-MYC, an important transcription factor for pluripotent stem-cell metabolism [53,54], as well as for metabolic reprogramming in cancer cells, mediates the expression of glycolytic genes [53,54], such as *ldha* [53]. Therefore, the decreased expression of the c-MYC protein might be influencing this downregulation in the absence of Leu and Arg. Nonetheless, it should also be noted that naïve embryonic cells display an ambivalent metabolism [24,25,55], with the ability to use both glycolysis and oxidative metabolism for ATP production. Interestingly, oligomycin injection only mildly accelerated the glycolysis rate in Leu- and Arg-starved cells. Oligomycin inhibits mitochondrial ATP synthesis, and upon its injection, cells are forced to fulfill their energetic demands for ATP production through glycolysis. Therefore, we assessed the overall oxidative profile, showing that it was also decreased in the absence of both amino acids. Additionally, cells were responsive to all the inhibitors used, suggesting that the mitochondria respiratory chain was not impaired, but its function was reduced, possibly due to lower energy demand. This could be explained by a scarcity in Krebs cycle intermediates, not only due to decreased pyruvate uptake and reduced glycolysis activity but also by lower levels of COXIV, important for Complex IV assembly [56] and therefore for mitochondrial activity. Although a more general and indirect effect of Leu and Arg absence could be postulated (in terms of, for example, lower synthesis of transporters, enzymes involved in metabolic pathways, or DNA synthesis and transcription) this would likely also affect the cells globally and result in more cell death or changes in the pluripotency network or in cell differentiation, which we did not detect. There is a fine balance between metabolic switches and differentiation [55,57,58,59,60,61] but, although the results suggest a metabolic downregulation, pluripotency was never affected in all our experimental conditions.

Metabolic modulation has been regarded as an easier, less harsh and powerful tool for regulation cellular fate than pharmacological approaches. The data presented show that the mere removal of Leu and Arg from the culture medium can induce a quiescent state in naïve mouse embryonic stem cells. Amino acid composition of the uterine fluid was shown to be crucial for diapause, and our results support this notion. It would also be interesting to analyze how these amino acids contribute to mESC regulation and epigenetic signature. Future studies focused on the functional and molecular routes of these amino acids will be promising on the comprehension of quiescence regulation in many cell types, which still remains a black box to some extent. This will uncover many promising methods to improve aging-associated pathological conditions, for example, also allowing for a better understanding of molecular regulation not only of diapause and paused-pluripotency, as well as those related to other types of stem cells, such as cancer-stem cells.

## 4. Materials and Methods

### 4.1. Cell Culture and Maintenance

Mouse embryonic stem cells (E14TG2a) were grown on 0.1% gelatin (Sigma-Aldrich, St. Louis, MO, USA)-coated culture dishes at 37 °C and in a 5% CO_2_ atmosphere. Cells were cultured in N2B27-based medium composed by DMEM-F12 (Gibco Invitrogen, Waltham, MA, USA) and Neurobasal (Gibco Invitrogen, Waltham, MA, USA) in a 1:1 ratio, supplemented with 100 U/mL Penicillin/Streptomycin (Gibco Invitrogen, Waltham, MA, USA), 0.75 mM L-glutamine (Gibco Invitrogen, Waltham, MA, USA), 0.1 mM 2-mercaptoethanol (Sigma-Aldrich, St. Louis, MO, USA), 1x B27 supplement (Gibco Invitrogen, Waltham, MA, USA), 0.5x N2 supplement (Gibco Invitrogen, Waltham, MA, USA), 1 × 10^3^ U/mL of Leukemia Inhibitory Factor (Gibco Invitrogen, Waltham, MA, USA), and 1 μM PD0325901 (Axon Medchem, Groningen, The Netherlands), 3 μM CHIR99021 (Axon Medchem, Groningen, The Netherlands). This media will be referred as 2i/LIF media.

For passaging, colonies were dissociated with StemPro™ Accutase™ Cell Dissociation Reagent (Merck-Millipore, Darmstadt, Germany). Cells were then centrifuged at 300× *g* for 5 min, resuspended in fresh media witb a single-cell suspension, and counted with trypan blue (Sigma-Aldrich, St. Louis, MO, USA) in a Neubauer chamber, before being plated in 0.1% of gelatin-coated culture dishes at a chosen density. For experimental conditions, cells were plated at 8000 cells/cm^2^ density in 2i/LIF medium to allow them to properly adhere. After 12 h, media was removed, cells were rinsed with 1x PBS, and fresh media for each experimental condition was added. To culture cells in the absence of key amino-acids (xAA condition), cells were cultured on 2i/LIF medium with customized DMEM-F12 (Gibco Invitrogen, Waltham, MA, USA) and Neurobasal (Gibco Invitrogen, Waltham, MA, USA) media without both L-Leucine and L-Arginine. xAAxLIF cells were cultured with xAA medium without LIF supplementation. xLIF cells were cultured in 2i media in the absence of LIF.

In the recovery experiments, after 48 h of culture with the different treatments, the media used for each condition was removed, and cells were rinsed with 1x PBS. Afterwards, for 2i/LIF and xLIF, fresh 2i/LIF media was added. In the xAA and xAAxLIF conditions, the medium added was the 2i/LIF medium without L-Leucine and L-Arginine supplemented with L-Leucine (Sigma-Aldrich, St. Louis, MO, USA) and L-Arginine (Sigma-Aldrich, St. Louis, MO, USA), matching the final concentration found in DMEM-F12/neurobasal media (82 mg/L and 115.75 mg/L, respectively).

### 4.2. Cell Proliferation Assay

Cells were plated in 48-well plates and counted every 24 h. For this purpose, cell medium was removed, and cells were rinsed with 1x PBS, dissociated with StemPro™ Accutase™ Cell Dissociation Reagent (Merck-Millipore, Darmstadt, Germany), and then inactivated by dilution with medium in order to obtain a cell suspension in an appropriated final volume. Afterwards, 10 μL of cell suspension was mixed with 10 μL of Trypan Blue (Sigma-Aldrich, St. Louis, MO, USA) and counted in a Neubauer chamber. In the long-term culture assays, in L-Leucine and L-Arginine absence conditions, from day 6 of culture, cells were centrifuged after dissociation and resuspended in an appropriated final volume to allow for counting.

Growth rates (***GR***) were calculated using the proliferation assay data and the following formula, where ***N*** is number of cells in ***t*** time, and ***N*0** is the number of cells in ***t*0** time:(1)$$GR=\frac{\ln\left(N\right)-\ln\left(N0\right)}{t-t0}$$

### 4.3. Flow Cytometry

Cell cycle, apoptosis by Annexin V/PI staining, and nascent RNA synthesis were evaluated by flow cytometry using a BD FACSCalibur cytometer, with data acquisition and analysis performed in CellQuest Software (BD Biosciences, Heidelberg, Germany).

For cell-cycle analysis, cells were treated according to the experimental design, and every 24 h, cells were harvested and fixed with ice-cold ethanol (70%), to be later kept at −20 °C for at least 24 h. Fixed cells were washed twice with ice cold 1x PBS and then incubated at room temperature (RT) in the dark with a propidium iodide (PI) solution composed of 1x PBS, 5 μM PI (Life Technologies, Waltham, MA, USA), 0.1% Triton x-100 (Sigma-Aldrich, St. Louis, MO, USA), and 0.1 mg/mL RNAse (Invitrogen, Waltham, MA, USA). After 30 min of incubation, cells were centrifuged at 350× *g* for 10 min, resuspended in ice-cold 1x PBS, and analyzed in the cytometer, whereby a total 20,000 events were acquired. Unstained cells were used as a negative control.

Annexin V/PI staining was performed using the Annexin V (Immunostep, Salamanca, Spain) and PI protocol [62]. Cells were plated according to the experimental design for 48 h, then dissociated with StemPro™ Accutase™ Cell Dissociation Reagent (Merck-Millipore, Darmstadt, Germany) and counted. A total of 1 × 10^6^ live cells from each condition were then incubated with 5 μL annexin V-FITC (Immunostep, Salamanca, Spain) and 2.5 μM PI, both diluted in 1x annexin-binding buffer (Immunostep, Salamanca, Spain) for 15 min and analyzed in the cytometer. Unstained live cells were used as a negative control, and as a positive control, cells were previously incubated with 0.25% H_2_O_2_ (Sigma-Aldrich, St. Louis, MO, USA).

For nascent RNA synthesis analysis, the Click-iT^®^ RNA labeling kit (Cat.No:C10330) (Thermo Fisher Scientific, Waltham, MA, USA) was used. Cells were plated and cultured for 48 h in each experimental condition and then incubated for 1 h in the respective media supplemented with 1 mM 5-ethynyl uridine (EU) (Thermo Fisher Scientific, Waltham, MA, USA) at 37 °C and a 5% CO_2_ atmosphere. An extra dish of cell culture in 2i/LIF was previously incubated with 10µg/mL of actinomycin (ACTD) (Thermo Fisher Scientific, Waltham, MA, USA) for 3 h at 37 °C and a 5% CO_2_ atmosphere to be used as a negative control for nascent RNA synthesis. Following EU incubation, cells were detached, centrifuged, rinsed with 1x PBS, and fixed with 4% paraformaldehyde for 15 min at RT, then permeabilized in a 0.5% Triton-X100 PBS solution for an additional 15 min at RT. Permeabilized cells were incubated with the Click-iT^®^ RNA reaction cocktail (Thermo Fisher Scientific, Waltham, MA, USA), containing Alexa Fluor^®^ 647 picolyl azide (Cat.No: A10277, Thermo Fisher Scientific, Waltham, MA, USA), for 30 min at RT in the dark. After that, cells were rinsed with 1x PBS, and after centrifugation, the pellets were resuspended in 1x PBS, prior to flow cytometry analysis.

### 4.4. Immunocytochemistry and Protein Translation Assay

For immune detection of OCT4, cells were fixed with 4% of paraformaldehyde (Sigma-Aldrich, St. Louis, MO, USA) for 15 min at RT, permeabilized with ice-cold 100% methanol (Sigma-Aldrich, St. Louis, MO, USA) for 10 min at −20 °C, and blocked for 1 h in PBS containing 1% Triton X-100 and 5% BSA (Sigma-Aldrich, St. Louis, MO, USA). A solution of PBS containing 1% BSA and 0.25% Triton X-100 was used to dilute antibodies. The primary antibodies rabbit anti-Oct4 antibody (Cat.No:701756; Thermo Fisher Scientific, Waltham, MA, USA, 1:250), anti-alpha smooth muscle actin (Cat.No: ab7817, Abcam, Cambridge, UK; 1:150), anti-alpha 1 Fetoprotein (Cat.No: ab213328, Abcam, Cambridge, UK; 1:50), anti-b-Tubulin Isotype III (Cat.No:T5076 Sigma, St. Louis, MO, USA; 1:150), and NANOG (Cat.No:ab80892, Abcam, Cambridge, UK; 1:200) were incubated overnight at 4 °C. Texas Red-X goat anti-rabbit IgG (Cat.No:T6391, Thermo Fisher Scientific, Waltham, MA, USA) secondary antibodies were incubated for 45 min at RT in the dark, followed by a 10 min incubation with Hoechst 33,342 Trihydrochloride (Invitrogen, Waltham, MA, USA).

For nascent protein synthesis analysis, the Click-iT^®^ Plus OPP Protein Synthesis Assay Kit (Cat.No:C10457) (Thermo Fisher Scientific, Waltham, MA, USA) was used. Cells were plated on a IBIDI µ-Slide 8 well, cultured for 48 h in each experimental condition, and then incubated for 1 h in the respective media supplemented with 50 µM O-propargylpuromycin (OP-Puro) (Thermo Fisher Scientific, Waltham, MA, USA) at 37 °C and a 5% CO_2_ atmosphere. An extra well of cells cultured in 2i/LIF was previously incubated with 50 µM of cyclohexamide (CHX) (Abcam, Cambridge, UK), for 1 h at 37 °C and a 5% CO_2_ atmosphere, to be used as a negative control for nascent protein synthesis. Following OP-Puro incubation, cells were fixed with 4% paraformaldehyde for 15 min at RT, then permeabilized in a 0.5% Triton-X100 PBS solution for an additional 15 min at RT. Permeabilized cells were incubated with the Click-iT^®^ Plus OPP reaction cocktail (Thermo Fisher Scientific, Waltham, MA, USA) containing Alexa Fluor^®^ 594 picolyl azide (Thermo Fisher Scientific, Waltham, MA, USA), according to the manufacturers’ indications, for 30 min at RT in the dark. After that, cells were rinsed with Click-iT^®^ Reaction Rinse Buffer (Thermo Fisher Scientific, Waltham, MA, USA), and the nuclei were stained with 1XHCS NuclearMask™ Blue Stain working solution (Thermo Fisher Scientific, Waltham, MA, USA) for 30 min at RT. Imaging was performed by confocal microscopy with an LSM 710 Zeiss Confocal Microscope. Image analysis was performed using the Image J software.

### 4.5. RNA Isolation, DNA Cleanup, cDNA Synthesis, and Quantitative Real-Time PCR (qRT-PCR)

Nucleic acids were extracted after 48 h of treatment, after 24 h of recovery for all the experimental conditions, and after 48 h for only xAA and xAAxLIF. Cells from each condition were harvested, and nucleic acid total content was isolated using the TRIzol reagent (Invitrogen, Waltham, MA, USA), vortexed, and then mixed with 200 μL chloroform (Sigma-Aldrich, St. Louis, MO, USA) and vortexed again. Afterwards, samples were centrifuged at 12,000× *g* for 15 min at 4 °C to separate the two phases, the TRIzol protein phase and a chloroform aqueous phase containing nucleic acids that was collected, mixed with 2-propanol (Sigma-Aldrich, St. Louis, MO, USA) on a 1:1 ratio to precipitate RNA, and stored at −20 °C. Before DNA cleanup, samples were centrifuged, washed in 600 μL 75% ethanol, and left to air dry for 5–10 min. Then, pellets were dissolved in nuclease-free water. DNA was eliminated using the DNA-free kit (Ambion, Invitrogen, Waltham, MA, USA). Next, samples were centrifuged, and RNA concentration and quality were determined in a NanoDrop 2000 (Thermo Fisher Scientific, Waltham, MA, USA). Finally, only samples with a 260/280 ration at least equal to 1.8 were stored at −80 °C until usage. After extraction and DNA cleanup, 1 μg of RNA was converted in cDNA in 0.2 mL PCR tubes using the iScript cDNA Synthesis Kit (BioRad, Hercules, CA, USA), according to the manufacturer’s protocol. The reaction took place in a C1000^TM^ Thermal Cycler.

For RT-PCR experiments, mouse-specific primers from a primer bank database (http://pga.mgh.harvard.edu/primerbank/) were used (accessed on 25 October 2019). *Oct4*-forward primer (FP): GGCTTCAGACTTCGCCTCC/reverse primer (RP):AACCTGAGGTCCACAGTATGC; *Nanog* –FP:TCTTCCTGGTCCCCACAGTTT/RP:GCAAGAATAGTTCTCGGGATGAA; *Rex1* –FP:CCCTCGACAGACTGACCCTAA/RP:TCGGGGCTAATCTCACTTTCAT; *Esrrb* –FP:GCACCTGGGCTCTAGTTGC/RP:TACAGTCCTCGTAGCTCTTGC; *Hxk2*—FP:TGATCGCCTGCTTATTCACGG/RP:AACCGCCTAGAAATCTCCAGA; *Pdh*—FP:GAAATGTGACCTTCATCGGCT/RP:TGATCCGCCTTTAGCTCCATC; *Pkm*—FP:GCCGCCTGGACATTGACTC/RP:CCATGAGAGAAATTCAGCCGAG; *Ldha*—FP:TGTCTCCAGCAAAGACTACTGT/RP:GACTGTACTTGACAATGTTGGGA; *Cox*—FP:GCGTCTGCGGGTTCATATTG/RP:TCTGCATACGCCTTCTTTCTTG; *Pdhk1*—FP:GGACTTCGGGTCAGTGAATGC/RP:TCCTGAGAAGATTGTCGGGGA; *Sdha*—FP:GGAACACTCCAAAAACAGACCT/RP:CCACCACTGGGTATTGAGTAGAA; *Beta-Actin(Actb)*—;FP:GGCTGTATTCCCCTCCATCG/RP:CCAGTTGGTAACAATGCCATGT; Afp—FP: CTTCCCTCATCCTCCTGCTAC/RP: ACAAACTGGGTAAAGGTGATGG; Sma—FP: GTCCCAGACATCAGGGAGTAA/RP: TCGGATACTTCAGCGTCAGGA; Tubb3—FP: TAGACCCCAGCGGCAACTAT/RP: GTTCCAGGTTCCAAGTCCACC.

SsoFast EvaGreen Supermix (Bio-Rad, Hercules, CA, USA) was used to perform the RT-PCR reaction, and the analysis was performed according to Bio-Rad instructions in a CFX96 Touch^TM^ Real-Time PCR Detection System. Gene expression was calculated using the -∆∆Ct method and normalized to the housekeeping gene ß-Actin.

### 4.6. Protein Extraction and Quantification and Western Blotting

For protein extraction, cells were lysed for 5 min on ice after vortex, with 100 μL of RIPA buffer (Sigma-Aldrich, St. Louis, MO, USA) supplemented with 2 mM PMSF (phenylmethylsulphonyl fluoride Sigma Aldrich, St. Louis, MO, USA), 2x Halt phosphatase inhibitor cocktail (Pierce, Rockford, IL, USA; Thermo Fisher), and protease inhibitor cocktail CLAP (Sigma-Aldrich, St. Louis, MO, USA). Protein was quantified in duplicates using the PierceTM BCA (Bicinchonic Acid) Protein Assay Kit (Thermo Fisher Scientific, Waltham, MA, USA) following the manufacturer’s protocol, and absorbance was determined in a BioTek Synergy HT multi-detection microplate reader (BioTek Instruments, Winooski, VT, USA). An amount of 30 μg of protein extract was denaturated using a Laemmeli sample buffer (Bio-Rad, Hercules, CA, USA) for 10 min at 75 °C and then loaded in 7.5% or 12% Acrilamide Tris-HCl gels. Eletrophoresis was carried out in a Mini Protean Tetra Cell (Bio-Rad, Hercules, CA, USA) system, and then proteins were electrotransfered to polyvinylidene difluoride (PVDF) membranes (Bio-Rad, Hercules, CA, USA). Membranes were blocked for 1 h RT in 5% BSA or 5% milk diluted in Tris—Buffered Saline with Tween (TBS-T) composed of 5 mM Tris, 15 mM NaCl (Sigma-Aldrich, St. Louis, MO, USA), and 0.1% of Tween 20 (Sigma-Aldrich, St. Louis, MO, USA). Then, they were incubated at 4 °C overnight with the following primary antibodies: rabbit anti-NANOG (Cat.No:ab80892, Abcam, Cambridge, UK; 1:500), rabbit anti-OCT4 (Cat.No:701756, Thermo Fisher Scientific, Waltham, MA, USA, 1:250), rabbit anti-SOX2 (Cat.No:ab97959, Abcam, Cambridge, UK; 1:900), rabbit anti-c-MYC (Cat.No:#9402, Cell Signaling, Danvers, MA, USA; 1:1000),rabbit anti-PKM2 (Cat.No:ab137852, Abcam, Cambridge, UK; 1:250), mouse anti-HXK II (Cat.No:sc-130358, Santa Cruz, Dallas, TX, USA; 1:200), mouse anti-PDH-E1alpha (Cat.No:sc-377092, Santa Cruz, Dallas, TX, USA; 1:100), mouse anti-COX IV (Cat.No:#11967; Cell Signaling, Danvers, MA, USA, 1:1000), rabbit anti-SDHA (Cat.No:#11998; Cell Signaling, Danvers, MA, USA, 1:1000), rabbit anti-PDHK1 (Cat.No:#3820, Cell Signaling, Danvers, MA, USA; 1:1000), rabbit anti-p-LDHA (Tyr10) (Cat.No:#8176, Cell Signaling, Danvers, MA, USA; 1:1000), rabbit anti-LDHA (Cat.No:PA5-27406; Thermo Fisher Scientific, Waltham, MA, USA, 1:500), and goat anti-CANX (Cat.No:ab0041, SICGEN, Cantanhede, PT; 1:2500). Secondary antibodies,were diluted in 5% BSA in TBS-T, except for Calnexin and anti-goat antibodies, which were diluted in 5% and 2.5% milk in TBS-T (Bio-Rad, Hercules, CA, USA), respectively. A 1:1 solution of enhancer reagent and oxidizing reagent of either ImmunoStar ECL substrate (Bio-Rad, Hercules, CA, USA) or Western BrightTM Sirius (Advansta, San Jose, CA, USA) were used to detect secondary antibodies’ signals. Membranes were developed using ImagQuant LAS 500 (GE Healthcare Bio-Sciences AB, Uppsala, Sweden), data were acquired using Quantity One software, and signal densitometry was quantified using ImageJ 1.52a software (US National Institutes of Health). All proteins were normalized to Calnexin.

### 4.7. Quantification of Extracellular Metabolites in Media

Media was filtered with 0.2 μm filters (GE Healthcare Life Sciences, Chicago, IL, USA). For pyruvate detection, media was deproteinized with 8% perchloric acid (Sigma-Adrich, St. Louis, MO, USA) on a 2:3 ratio, and supernatants were collected after centrifugation. Filtered and deproteinized media were kept at −80 °C. All the metabolite quantification techniques were kindly performed by Dr. Inês Baldeiras in the Laboratory of Neurochemistry at the University Hospital of Coimbra (HUC) Neurology Service.

All the compounds were subjected to enzymatic methods, and detection was based on their stochiometric reactions using High Performance Liquid Chromatography (HPLC) quantification. Since pyruvate is converted to lactate in the presence of NADH (Roche, Basel, Switzerland) by the Lactate Dehydrogenase (Roche, Basel, Switzerland), pyruvate was dosed by measuring the decrease in NADH absorbance at 340 nm. Glucose was dosed by the Glycose Oxidase (GOD)-Peroxidase (POD)-colorimetric method (Randox, Crumlin, UK) at a 546 nm wavelength. Lactate was determined using the Randox Lactate assay (Randox, Crumlin, UK) by first being converted by lactate-oxidase enzyme to pyruvate and H_2_O_2,_ which in turn was converted to a colored product by Peroxidase detected at 550 nm.

### 4.8. Embryoid Body Assay

The differentiation capacity of mESCs cultured in the different conditions was evaluated using the embryoid bodies (EBs) suspension protocol [63]. For this assay, cells must be maintained in suspension; therefore, 10^6^ mESCs cultured in each condition for 48 h were plated in non-adherent 60 mm Petri dish, in regular 2i medium without LIF supplementation, for three days. Medium was changed using the EB sedimentation technique [63]. After three days in suspension, EBs were plated in a 100 mm tissue-cultured Petri dish and in 24 multiwell plates, coated with 0.1% gelatin to allow adherence, in a 1:1 DMEM-F12/neurobasal medium supplemented with 25% FBS and 100 U/mL penicillin/streptomycin. The medium was changed every day. Protein and RNA were extracted after 15 days from the protocol initiation, and cells were plated in multiwell plates fixed for immunocytochemistry assays.

### 4.9. Live-Cell Metabolic Assays

A total of 100,000 cells were plated 12 h before the assay for each experimental condition in duplicates in the 24-well XF24 cell culture plate and maintained in their specific media according to the experimental conditions. One hour before the assay, cells were rinsed with XF Glycolysis-specific media and incubated in this media for 1 h at 37 °C with no CO_2_. For extracellular acidification rate (ECAR) measurement, glycolytic function XF Assay Medium Modified DMEM (Agilent, Santa Clara, CA, USA) was supplemented with 200 μM L—glutamine with a pH = 7.4. For oxygen consumption rate (OCR) measurement, XF Assay Medium Modified DMEM (Agilent, Santa Clara, CA, USA), adjusted at pH = 7.4, was supplemented with 4.5 g/l glucose, 2 mM pyruvate, and 200 μM L-glutamine. To measure glycolytic function, Seahorse XF Glycolysis Stress Kit compounds (Agilent, Santa Clara, CA, USA) glucose (10 mM), Oligomycin (1 μM), and 2-deoxyglucose (100 mM) were loaded to the XFe24 Sensor Cartridge and injected after measurements 3, 6, and 9, respectively. To assess mitochondrial function, oligomycin (1 μM), FCCP (1.25 μM), and rotenone plus antimycin A (1 μM each) were loaded to the XFe24 Sensor Cartridge and injected after measurements 3, 6, and 9, respectively. After all measurements were completed, cells were dissociated and counted for Seahorse data normalization.

## 5. Statistical Analysis

Statistical analysis was executed using SPSS 21.0 (SPSS Inc., Chicago, IL, USA). Normality and homoscedasticity were analyzed using the Shapiro–Wilk and Levene tests, respectively. A one-way ANOVA or independent t-Student test were performed whenever data respected the normal distribution, followed by Bonferroni or Dunnett T3 post-hoc tests (after one-way ANOVA test) to determine statistical significances, considering the data homoscedasticity. If normality was not verified the Kruskal–Wallis non-parametric test, the Mann-Whitney was performed test to determine statistical significances. The threshold for statistical significances was *p* ≤ 0.05. Data are expressed as means ± standard error of mean (SEM). Statistical significance is displayed as * *p* < 0.05, ** *p* < 0.01, *** *p* < 0.001.

## Figures and Tables

**Figure 1 ijms-23-14286-f001:**
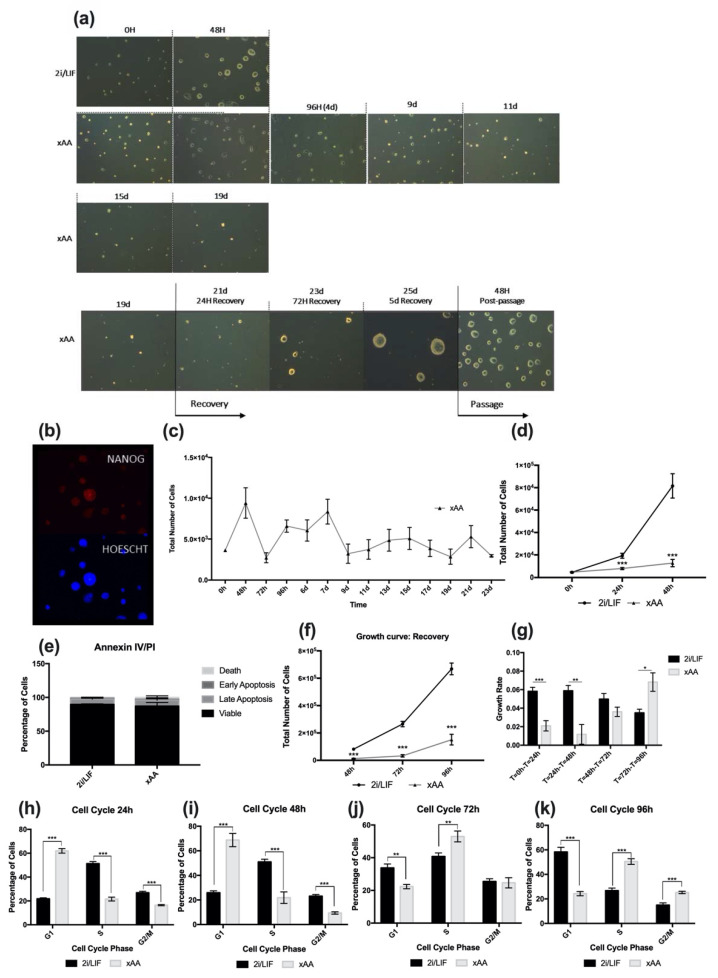
**2i mESCs can be maintained in long-term culture in the absence of Leucine and Arginine.** (**a**) Representative images acquired from randomly selected fields using phase contrast microscopy (100× magnification). Images include: 1—cells cultured in 2i/LIF medium for 48 h (control, first row); 2—cells cultured in the absence of Leu and Arg (xAA) for 19 days, with images taken at several time points (second and third rows); 3—cells cultured in the absence of Leu and Arg in xAA medium for 19 days, followed by replacement with regular 2i/LIF medium and culture for a further 5 days (recovery); and 48 h after passaging recovered cells in regular 2i/LIF medium (fourth row). (**b**) Representative fluorescence microscopy images of colonies 48 h after passaging xAA cells cultured in Leu and Arg absence for 20 days and recovered with fresh 2i/LIF media for 5 days, stained for NANOG (magnification 630×). (**c**) Growth curve of mESCs cultured in 2i/LIF media in the absence of Leu and Arg (xAA) for 23 days. Results are presented as mean ± SEM, (*n* = 3/group). (**d**) Proliferation of naive mESCs cultured in 2i/LIF media and of Leu and Arg (xAA). Cells were counted every 24 h until the 48 h time point. Results are presented as mean ± SEM, (2i/LIF: *n* = 10) and (xAA: *n* = 8). (**e**) After 48 h of culture in 2i/LIF and xAA conditions, cells were incubated with Annexin V and PI and analyzed by flow cytometry. Results depicting the percentage of cells gated for each quadrant are presented as mean ± SEM (*n* = 4/group). (**f**) Proliferation during recovery of naive mESCs previously cultured in 2i/LIF media and xAA culture conditions, after which the regular culture conditions were reestablished for the xAA condition (see Section 4). Cells were counted every 24 h until the 96 h time point (48 h of recovery). Results are presented as mean ± SEM (2i/LIF: *n* = 10) and (xAA: *n* = 8). (**g**) Growth rate of naive mESCs, calculated from the proliferation data represented in (**d**) and (**f**). See Section 4 for calculations. Results are presented as mean ± SEM (2i/LIF: *n* = 10); (xAA: *n* = 8). (**h**) Cell-cycle analysis of mESCs after 24 h of culture in 2i/LIF and xAA culture conditions. Cells were detached, treated with propidium iodide, and analyzed by flow cytometry. The percentage of cells in G1, S, and G2/M phases of the cell cycle were analyzed in each culture condition. Results are presented as mean ± SEM, (2i/LIF 24 h: *n* = 6/phase); (xAA: *n* = 6/phase). (**i**) Cell-cycle analysis of mESCs after 48 h of culture in 2i/LIF and xAA culture conditions. Results are presented as mean ± SEM, (2i/LIF 48 h: *n* = 9/phase); (xAA: *n* = 9/phase). (**j**) Cell-cycle analysis of mESC recovery at the 72 h time point, when the regular culture conditions were reestablished (see Section 4). Cells were previously cultured in each experimental condition, then detached, treated with propidium iodide, and analyzed by flow cytometry. The percentage of cells in G1, S, and G2/M phases of the cell cycle was analyzed in each culture condition. Results are presented as mean ± SEM, (2i/LIF 72 h: *n* = 9/phase); (xAA 72 h: *n* = 9/phase). (**k**) Cell-cycle analysis of mESC recovery, 96 h time point, when the regular culture conditions were reestablished (see Section 4). Results are presented as mean ± SEM, (2i/LIF 96 h: *n* = 8/phase); (xAA 96 h: *n* = 8/phase). Statistical significance considered when * *p* < 0.05, ** *p* < 0.01, and *** *p* < 0.001. Specific statistical tests: a *t*-test was performed for the T0 h, T48 h, T72 h, and T96 h of the growth curves; late apoptosis and dead cells in Annexin IV/PI; growth rate; cell cycle 24 h, G2 phase of cell cycle at 48 h, cell cycle 72 h, cell cycle 96 h. A Mann–Whitney test was performed for the T24 h of the growth curve; early apoptosis and viable cells in Annexin IV/PI; G1 and S phases of cell cycle 48 h.

**Figure 2 ijms-23-14286-f002:**
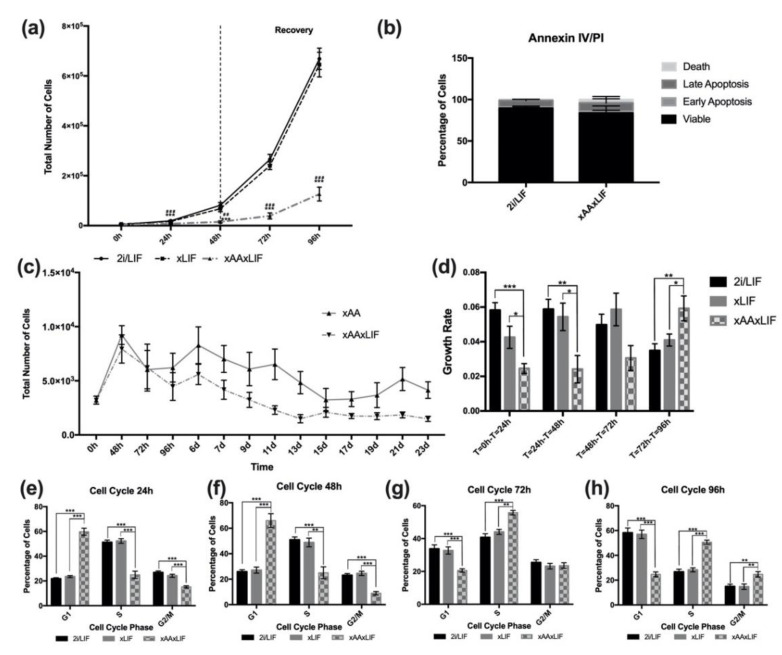
**Effects of LIF and Leucine and Arginine withdrawal on mESC proliferation and cell cycle.** (**a**) Proliferation of naive mESCs cultured in 2i/LIF media, in the absence of LIF (xLIF) and in the absence of LIF, Leu, and Arg (xAAxLIF). Cells were counted every 24 h until the 96 h time point. At the 48 h time point, the regular culture conditions were reestablished for xLIF and xAAxLIF conditions for the recovery period. The * symbol represents the statistical differences between 2i/LIF and xAAxLIF; the # symbol represents the statistical differences between xLIF and xAAxLIF. Results are presented as mean ± SEM, (2i/LIF: *n* = 10); (xLIF: *n* = 8); (xAAxLIF: *n* = 8), (**b**) After 48 h of culture in 2i/LIF and xAAxLIF conditions, cells were incubated with Annexin V and PI and analyzed by flow cytometry. Results depicting the percentage of cells gated for each quad are presented as mean ± SEM (*n* = 4/group). (**c**) Growth curve of mESCs cultured in 2i/LIF media in the absence of Leu and Arg (xAA) (Figure 1c), in the absence of LIF, Leu, and Arg (xAAxLIF) for 23 days. Results are presented as mean ± SEM, *n* = 3/group. (**d**) Growth rate of n mESCs, calculated from the proliferation data represented in Figure 1d,f and Figure 2a. See Section 4.2 for calculations. Results are presented as mean ± SEM (2i/LIF: *n* = 10); (xLIF: *n* = 8); (xAA: *n* = 8); (xAAxLIF: *n* = 8). (**e**) Cell-cycle analysis of mESCs after 24 h of culture in 2i/LIF, xLIF, and xAAxLIF culture conditions. Cells were detached, treated with propidium iodide, and analyzed by flow cytometry. The percentage of cells in G1, S, and G2/M phases of the cell cycle were analyzed in each culture condition. Results are presented as mean ± SEM, (2i/LIF 24 h: *n* = 6/phase); (xLIF: *n* = 6/phase); (xAAxLIF 24 h: *n* = 6/phase). (**f**) Cell-cycle analysis of mESCs after 48 h of culture in 2i/LIF and xAA culture conditions. Results are presented as mean ± SEM, (2i/LIF 48 h: *n* = 9/phase); (xAA: *n* = 9/phase). (**g**) Cell-cycle analysis of mESC recovery, at the 72 h time point, when the regular culture conditions were reestablished for the xLIF and xAAxLIF conditions (see Section 4.1). Cells were previously cultured in each experimental condition, then detached, treated with propidium iodide, and analyzed by flow cytometry. The percentage of cells in G1, S, and G2/M phases of the cell cycle was analyzed in each culture condition. Results are presented as mean ± SEM, (2i/LIF 72 h: *n* = 9/phase); (xLIF 72 h: *n* = 9/phase); (xAAxLIF 72 h: *n* = 9/phase). (**h**) Cell-cycle analysis of mESC recovery, 96 h time point, when the regular culture conditions were reestablished (see Section 4). Results are presented as mean ± SEM, (2i/LIF 96 h: *n* = 8/phase); (xLIF 96 h: *n* = 8/phase); (xAAxLIF 96 h: *n* = 8/phase). Statistical significance considered when * *p* < 0.05, ** *p* < 0.01 and *** *p* < 0.001. Specific tests: a one-way ANOVA followed by a Bonferroni or Dunnett T3 post-hoc tests, according to the homoscedasticity results, was performed for the T48 h of the growth curves; T = 0 h–T24 h, T = 48 h–T = 72 h, T = 72 h–T = 96 h of the growth rate; cell cycle 24 h; G2 phase of cell cycle at 48 h; cell cycle 72 h; cell cycle 96 h. A *t*-test was performed at T0 h of the growth curve for the 2i/LIF and xAAxLIF conditions, as well as for the G1 and S phases of cell cycle 48 h for the same conditions. A Mann–Whitney test was performed to compare the 2i/LIF and xLIF conditions to xAAxLIF. A Kruskal–Wallis test followed by a Mann–Whitney test (when applicable) were performed for T72 h of the growth curve; Annexin IV/PI; T24 h-T = 48 h of the growth rate.

**Figure 3 ijms-23-14286-f003:**
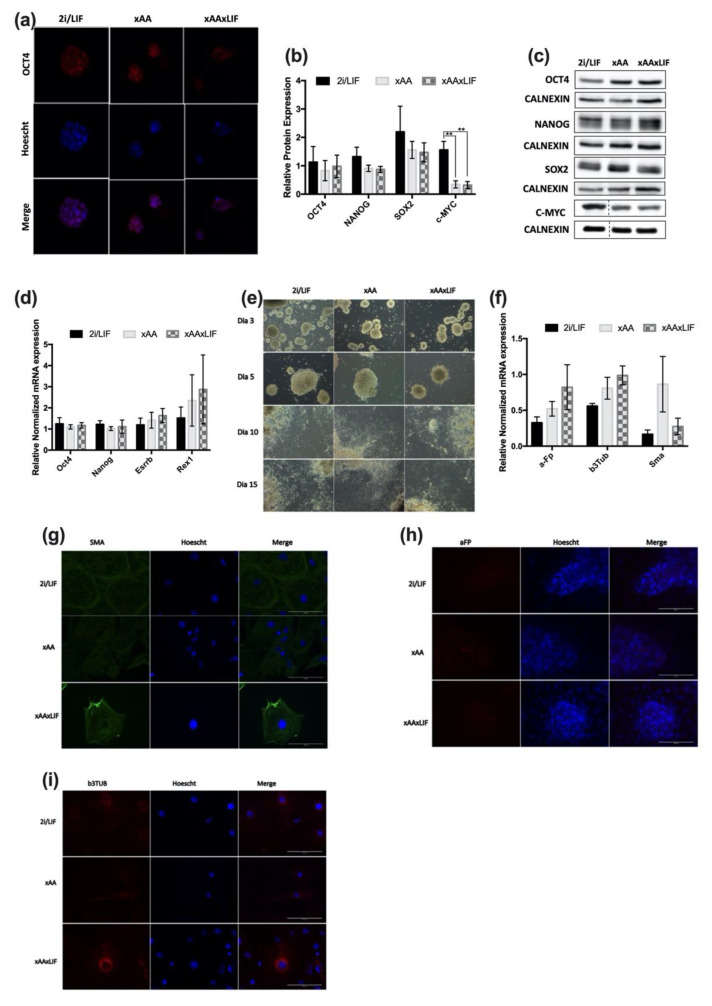
**Pluripotency is not affected by Leucine and Arginine withdrawal**. (**a**) Panel of fluorescence microscopy images of colonies of each of experimental condition: 2i/LIF, xAA, and xAAxLIF stained for OCT4. The representative images of the colonies for each group of conditions were obtained after 48 h of culture (magnification 630×). (**b**) OCT4, NANOG, SOX2, and c-MYC protein levels were determined by Western blot after 48 h in experimental culture conditions (2i/LIF, xAA, and xAAxLIF) and were normalized to CALNEXIN levels, used as loading control. Results are presented as mean ± SEM (OCT4: *n* = 5/protein/group; NANOG; SOX2 and c-MYC: *n* = 4/protein/group. (**c**) Representative immunoblot membranes for OCT4, NANOG, SOX2, and c-MYC and the relative CALNEXIN expression of each condition at 48 h. See Appendix A for the original membrane blot of C-MYC and the respective CALNEXIN. (**d**) Evaluation of pluripotency-related gene expression by RT-PCR. mRNA levels of *Esrrb*, *Nanog*, *Oct4,* and *Rex1* after 48 h in experimental culture conditions (2i/LIF, xAA and xAAxLIF). Each gene expression was normalized to *b-Actin* levels. Results are presented as mean ± SEM, (*Rex1*: n3/group; *Esrrb*, *Nanog*, *Oct4*: *n* = 4/gene/group). (**e**) Representative images acquired from randomly selected fields using phase-contrast microscopy (100× magnification) of EBs generated by cells that had been previously cultured in each experimental condition for 48 h before the differentiation assay. The images were acquired at day 3, 5, 10, and 15 of the assay. (**f**) Evaluation of anti-alpha 1 Fetoprotein (*a-Ft*) (endoderm marker), anti-b-Tubulin Isotype 3 (*b-3Tub*) (ectoderm marker), and anti-alpha smooth muscle Actin (*Sma*) (mesoderm marker) gene expression by RT-PCR at the day 15. Gene expression was normalized to *b-Actin* levels. Results are presented as mean ± SEM, (*n* = 4/group). (**g**) Panel of fluorescence microscopy images of EBs positive for the mesoderm marker SMA. EBs generated by cells that had been previously cultured in each experimental condition for 48 h before the differentiation assay. The representative images of EBs of each group of conditions were obtained at day 15 from the start of the assay (magnification 630×). (**h**) Panel of fluorescence microscopy images of EBs positive for the endoderm marker a-FT. EBs generated by cells that had been previously cultured in each experimental condition for 48 h before differentiation. The representative images of EBs of each group of conditions were obtained at day 15 from the start of the assay (magnification 630×). (**i**) Panel of fluorescence microscopy images of EBs positive for the ectoderm marker b-3TUB. EBs generated by cells that had been previously cultured in each experimental condition for 48 h before differentiation. The representative images of EBs of each group of conditions were obtained at day 15 from the start of the assay (magnification 630×). Statistical significance considered when ** *p* <0.01. Specific tests: a one-way ANOVA was performed for the mRNA expression results of the pluripotency markers; a-fp, b3tub, and sma markers; and for protein expression of NANOG and C-MYC. For C-MYC, the post-hoc test performed was Bonferroni. A Kruskal–Wallis test was performed for OCT4 and SOX2 protein expression.

**Figure 4 ijms-23-14286-f004:**
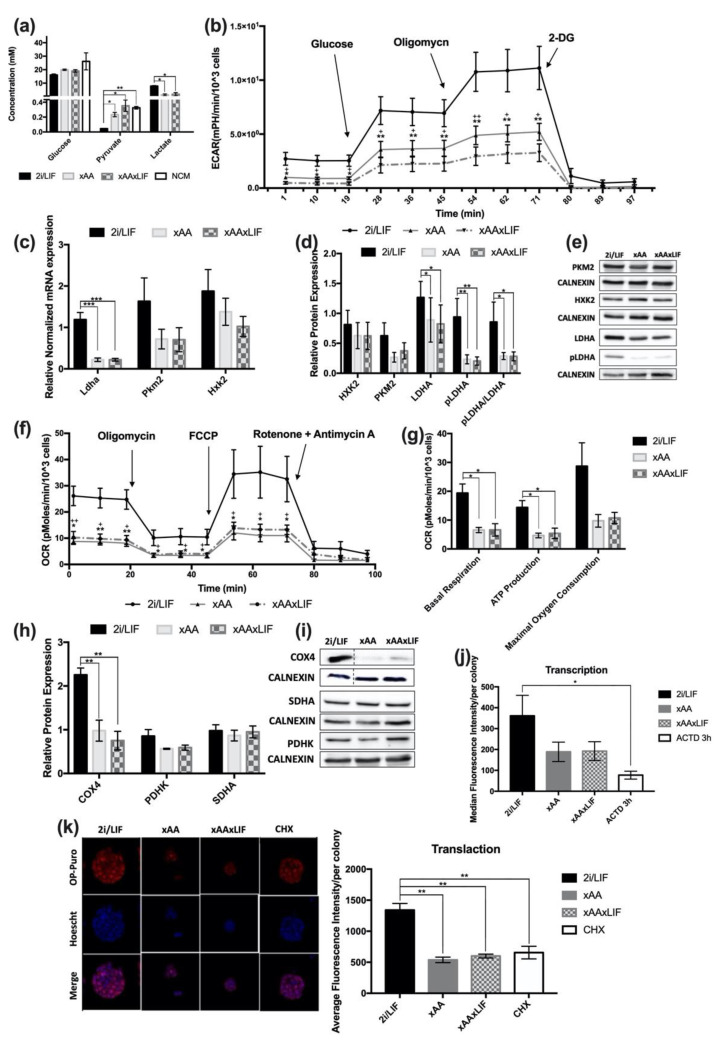
**Leucine and Arginine absence prompts differential nutrient uptake and reduces overall glycolytic function.** (**a**) Quantification of Glucose uptake from the extracellular medium after 48 h in experimental culture conditions (2i/LIF, xAA, and xAAxLIF) (NCM: *n* = 3); (2i/LIF, xAA, and xAAxLIF: *n* = 4). Pyruvate quantification after 48 h in all experimental culture conditions (2i/LIF, xAA, and xAAxLIF). (NCM: *n* = 3); (2i/LIF, xAA, and xAAxLIF: *n* = 4). Lactate quantification after 48 h in all experimental culture conditions (2i/LIF, xAA, and xAAxLIF), (NCM: *n* = 3) (2i/LIF, xAA, and xAAxLIF: *n* = 6). NCM-medium that was not in contact with cells. Results are presented as mean ± SEM. (**b**) Extracellular acidification rates (ECAR) measured using the Seahorse extracellular flux assay of cells maintained in 2i/LIF, xAA, and xAAxLIF conditions. Three measurements of ECAR were performed before and after the sequential injection of each of the three compounds: glucose (10 mM); oligomycin (1 μM), and 2-deoxyglucose (100 mM), respectively. The * symbol represents the statistical differences between 2i/LIF and xAAxLIF; the + symbol represents the statistical differences between 2i/LIF and xAA. Results are presented as mean ± SEM, (*n* = 4/group). (**c**) Evaluation of glycolytic-related gene expression by RT-PCR. mRNA levels of *Ldha*, *Pkm2,* and *Hxk2* after 48 h in experimental culture conditions (2i/LIF, xAA, and xAAxLIF). Each gene expression was normalized to *b-Actin* levels. Results are presented as mean ± SEM, (*n* = 4/gene/group). (**d**) HXK2, PKM2, and LDHA and phosphorylated-LDHA (Tyr10) protein levels were determined by western blot after 48 h in the different experimental culture conditions (2i/LIF, xAA, and xAAxLIF) and were normalized to CALNEXIN levels, used as a loading control. The pLDHA/LDHA ratio was calculated after the normalization of both targets to CALNEXIN. Results are presented as mean ± SEM (HXK2: *n* = 6/group; PKM2: *n* = 7/group; LDHA: *n* = 7/group; pLDHA: *n* = 6/group; pLDHA/LDHA: *n* = 6/group). (**e**) Representative immunoblot membrane images for HXK2, PKM2, LDHA, and pLDHA and the relative CALNEXIN expression of each condition at 48 h. (**f**) Cell oxygen consumption rate (OCR) measured by Seahorse extracellular flux assay in 2i/LIF, xAA, and xAAxLIF conditions. Three measurements of OCR were performed before and after the sequential injection of each compound: oligomycin (5 μM), FCCP (2.5 mM), and rotenone plus antimycin (2.5 mM), respectively. The * symbol represents the statistical differences between 2i/LIF and xAA; the + symbol represents the statistical differences between 2i/LIF and xAAxLIF. Results are presented as mean ± SEM, (2i/LIF: *n* = 6; xAA and xAAxLIF: *n* = 5). (**g**) Basal respiration, respiration rate associated with ATP production, and estimated maximal oxygen consumption related to OCR data are represented as mean ± SEM, (2i/LIF: *n* = 6; xAA and xAAxLIF: *n* = 5). (**h**) COXIV, PDHK, and SDHA protein levels were determined by western blot after 48 h in each experimental culture condition (2i/LIF, xAA, and xAAxLIF) and were normalized to CALNEXIN levels. Results are presented as mean ± SEM (SDHA: *n* = 7/group; COX4, PDHK: *n* = 4/group). (**i**) Representative immunoblot membranes images for COX4, PDHK, and SDHA and the relative CALNEXIN expression of each condition at 48 h. See Appendix A for the original membrane blot of COXIV and its respective CALNEXIN. (**j**) Panel of representative fluorescence microscopy images of OP-Puro incorporation for 1 h. Cells were previously cultured for 48 h in each of the 2i/LIF, xAA, and xAAxLIF conditions (magnification 630×)—(**left**). Average fluorescence intensity of OP-Puro incorporation for 1 h. Cells were previously cultured for 48 h in each of the 2i/LIF, xAA, and xAAxLIF conditions. Results are presented as mean ± SEM (*n* = 4/protein/group)—(**right**); (**k**) Median fluorescence intensity of cells positive for Alexa Fluor^®^ 594 after EU incorporation during 1 h. Cells were previously cultured for 48 h in 2i/LIF, xAA, and xAAxLIF conditions. Results are presented as median ± SEM (*n* = 6/group). Statistical significance considered when * *p* < 0.05, ** *p* < 0.01, and *** *p* < 0.001. Specific tests: a one-way ANOVA followed by a Bonferroni or Dunnett T3 post-hoc test (when applicable) were performed for glucose media quantification; ECAR measurements after glucose, oligomycin, and 2-DG injections; ldha, pkm2, and hkk2 mRNA expression; HXK2 protein expression; for the basal OCR measurements 2 and 3, OCR measurements after FCCP injection and 3rd point OCR measure after rotenone and antimycin A injection; for basal respiration, ATP production and maximal oxygen consumption; COX4 and SDHA protein expression; and average fluorescence intensity/colony of transcription. The Kruskal–Wallis test followed by Mann–Whitney test (when applicable) were performed for pyruvate and lactate media quantification; basal ECAR measurements; PKM2, LDHA, pLDHA, and pLDHA/LDHA protein expression; 1st point of basal OCR measure, OCR measurements 1 and 2 after rotenone and antimycin A injection and OCR measurements after oligomycin injection; for PDHK protein expression; median fluorescence intensity. A *t*-test was performed to compare the 2i/LIF and NCM conditions of the pyruvate media quantification.

## Data Availability

Not applicable.

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
