# Peer review of "Leucine and Arginine Availability Modulate Mouse Embryonic Stem Cell Proliferation and Metabolism"

_ijms, 2022, doi:10.3390/ijms232214286_

Round 1

Reviewer 1 Report

Authors report on mouse embryonic stem cell proliferation and metabolism in the absence and reintroduction of leucine and arginine. They well showed how proliferation and cell cycle are affected by Leucine and Arginine absence with and without LIF.

Clear statistical methods description and methodology.  

Some suggestions can be found below: 

17: Abstract: are not related, not not related

18: a lack of

25: a lack of

38: especially

57: the of effects? did you mean the side effect? the effects? the off-target effects?

74: into consideration

291 a lack of

293: there is, no the is

352: form, not from

402: acid play

489: Therefore, (space and comma) 

513: many, not may

742: , a parametric

743: extra space after dash in Mann-Whitney

Figure 3 plos in d and f do not include significance. 

Significance explanation not included in the legend of Figure 3.

Lwo resolution of figures when printed, especially barplots e, f, g and h in Figure 2.

Author Response

REVIEWER 1

We thank the Reviewer for the comments. Responses below.

Authors report on mouse embryonic stem cell proliferation and metabolism in the absence and reintroduction of leucine and arginine. They well showed how proliferation and cell cycle are affected by Leucine and Arginine absence with and without LIF.

Clear statistical methods description and methodology.  

We have added descriptions of the specific statistical methodologies used in each case in the Figure legends.

Some suggestions can be found below: 

17: Abstract: are not related, not not related

18: a lack of

25: a lack of

38: especially

57: the of effects? did you mean the side effect? the effects? the off-target effects?

74: into consideration

291 a lack of

293: there is, no the is

352: form, not from

402: acid play

489: Therefore, (space and comma) 

513: many, not may

742: , a parametric

743: extra space after dash in Mann-Whitney

 Corrections to all small changes noted above were carried out accordingly.

Figure 3 plos in d and f do not include significance. 

There are no statistically significant differences in those cases.

Significance explanation not included in the legend of Figure 3.

Added.

Low resolution of figures when printed, especially barplots e, f, g and h in Figure 2.

We have added better quality Figures to the submission (not embedded in the text) as per the Journals requirements.

Reviewer 2 Report

In their paper entitled “Leucine and Arginine availability modulate mouse embryonic stem cell proliferation and metabolism”, the Authors report that the absence of both Leucine and Arginine reduces stem cell proliferation through cell cycle arrest. However, this deficit does not affect pluripotency; in addition, the noticed effects are  reversible when both amino acids are added again to the culture media.

The paper is of interest, methods and results are well described, and the paper is worth of publication.

I have only a few minor comments:

1.      Abstract, line 17, and line 19: please write here the complete name of LIF (line 17) and mESCs (line 19);

2.      Introduction, line 46: please write here the complete meaning of ISC;

3.      Line 57: “never addressed the of effects Leu and Arg absence…” should be “never addressed the effects of Leu and Arg absence…”

4.      Legend to Figure 1, lines 114-116: the sentence “cells cultured in Leu and Arg 114 absence 2i/LIF medium (xAA) for 19 days, 5 days after media was replaced for 2i/LIF regular me-115 dium (recovery) and 48 hours after passaging recovered cells” is not completely clear; in addition, it should be probably better to write “cells cultured in 2i/LIF medium without Leu and Arg (xAA) for 19 days” instead of “cells cultured in Leu and Arg absence 2i/LIF medium (xAA) for 19 days”; as a whole, I suggest to briefly summarize the experimental sequence in order to make clear the meaning of the different pictures in the figure;

5.      As long as it concerns the data shown in Fig.4 (effects of Leucine/arginine absence on nutrient uptake and glycolysis rate), one possibility is that synthesis of transporters and glycolytic enzyme is decreased in the absence of the two amino acids, and that a decrease of these proteins then affects nutrient uptake and glycolysis); actually, the absence in the medium of an essential amino acid (leucine), and a semi-essential amino acid (arginine) can deeply influence synthesis of most proteins, such as enzymes involved in metabolic pathways, as well as enzymes involved in transcription and DNA synthesis; this simple explanation, in my opinion, should be discussed a bit more in the Discussion section.

Author Response

REVIEWER 2

We thank the Reviewer for the comments. Responses below.

In their paper entitled “Leucine and Arginine availability modulate mouse embryonic stem cell proliferation and metabolism”, the Authors report that the absence of both Leucine and Arginine reduces stem cell proliferation through cell cycle arrest. However, this deficit does not affect pluripotency; in addition, the noticed effects are reversible when both amino acids are added again to the culture media.

The paper is of interest, methods and results are well described, and the paper is worth of publication.

I have only a few minor comments:

  1. Abstract, line 17, and line 19: please write here the complete name of LIF (line 17) and mESCs (line 19);
  2. Introduction, line 46: please write here the complete meaning of ISC;
  3. Line 57: “never addressed the of effects Leu and Arg absence…” should be “never addressed the effects of Leu and Arg absence…”

The above corrections were made in the text

  1. Legend to Figure 1, lines 114-116: the sentence “cells cultured in Leu and Arg 114 absence 2i/LIF medium (xAA) for 19 days, 5 days after media was replaced for 2i/LIF regular me-115 dium (recovery) and 48 hours after passaging recovered cells” is not completely clear; in addition, it should be probably better to write “cells cultured in 2i/LIF medium without Leu and Arg (xAA) for 19 days” instead of “cells cultured in Leu and Arg absence 2i/LIF medium (xAA) for 19 days”; as a whole, I suggest to briefly summarize the experimental sequence in order to make clear the meaning of the different pictures in the figure;

We thank the Reviewer, and have clarified the experimental design.

  1. As long as it concerns the data shown in Fig.4 (effects of Leucine/arginine absence on nutrient uptake and glycolysis rate), one possibility is that synthesis of transporters and glycolytic enzyme is decreased in the absence of the two amino acids, and that a decrease of these proteins then affects nutrient uptake and glycolysis); actually, the absence in the medium of an essential amino acid (leucine), and a semi-essential amino acid (arginine) can deeply influence synthesis of most proteins, such as enzymes involved in metabolic pathways, as well as enzymes involved in transcription and DNA synthesis; this simple explanation, in my opinion, should be discussed a bit more in the Discussion section.

We thank the Reviewer for this suggestion, and included this possibility in the Discussion.